# A Comprehensive Study of Three Different Portable XRF Scanners to Assess the Soil Geochemistry of An Extensive Sample Dataset

**Ynse Declercq [1,2,]*** [ID], **Nele Delbecque [1]**, **Johan De Grave [3]**, **Philippe De Smedt [1]**, **Peter Finke [1]**, **Abdul M. Mouazen [1]**, **Said Nawar [1,4]**, **Dimitri Vandenberghe [3]**, **Marc Van Meirvenne [1]** **and Ann Verdoodt [1]**

[1]  Department of Environment, Ghent University, Coupure Links 653, 9000 Ghent, Belgium; Nele.Delbecque@UGent.be (N.D.); Philippe.DeSmedt@UGent.be (P.D.S.); Peter.Finke@UGent.be (P.F.); Abdul.Mouazen@UGent.be (A.M.M.); Said.Nawar@UGent.be (S.N.); Marc.VanMeirvenne@UGent.be (M.V.M.); Ann.Verdoodt@UGent.be (A.V.)

[2]  Laboratory of Environmental and Urban Ecology, Department of Bioscience Engineering, University of Antwerp, Groenenborgerlaan 171, 2020 Antwerp, Belgium

[3]  Department of Geology, Ghent University, Krijgslaan 281, 9000 Ghent, Belgium; Johan.DeGrave@UGent.be (J.D.G.); Dimitri.Vandenberghe@UGent.be (D.V.)

[4]  Soil and Water Department, Faculty of Agriculture, Suez Canal University, Ismailia 41522, Egypt

*  Correspondence: Ynse.Declercq@UGent.be

**Abstract:** The assessment of soil elemental concentrations nowadays mainly occurs through conventional laboratory analyses. However, proximal soil sensing (PSS) techniques such as X-ray fluorescence (XRF) spectrometry are proving to reduce analysis time and costs, and thus offer a worthy alternative to laboratory analyses. Moreover, XRF scanners are non-destructive and can be directly employed in the field. Although the use of XRF for soil elemental analysis is becoming widely accepted, most previous studies were limited to one scanner, a few samples, a few elements, or a non-diverse sample database. Here, an extensive and diverse soil database was used to compare the performance of three different XRF scanners with results obtained through conventional laboratory analyses. Scanners were used in benchtop mode with built-in soil calibrations to measure the concentrations of 15 elements. Although in many samples Cu, S, P, and Mg concentrations were up to 6, 12, 13, and 5 times overestimated by XRF, and empirical recalibration is recommended, all scanners produced acceptable results, even for lighter elements. Unexpectedly, XRF performance did not seem to depend on soil characteristics such as CaCO3 content. While performances will be worse when expanding to the field, our results show that XRF can easily be applied by non-experts to measure soil elemental concentrations reliably in widely different environments.

**Keywords:** benchtop portable X-ray fluorescence (XRF); spectroscopy; elemental analysis; soil; data quality

## 1. Introduction

X-ray fluorescence (XRF) spectrometry nowadays is granted much attention as an upcoming proximal soil sensing (PSS) technique. It has already shown to successfully estimate a whole suite of soil parameters in a number of disciplines, including pedology, archaeology, geology, agronomy, and soil pollution [1,2]. The analysis of soil elemental concentrations aids the assessment of soil mineralogy, fertility, salinity, acidity, granulometry, and cation exchange capacity (CEC), alongside metal uptake by plants and soil heavy metal contamination [1–6]. Compared to conventional laboratory

analysis, the determination of soil elemental concentrations by XRF is much faster, less expensive, and non-destructive. The best results are obtained when samples are homogenized, dried, finely ground, and sieved [1,7–13], but sample preparation is not always necessary as portable XRF scanners can be directly employed in the field.

Yet, XRF is still outshone by traditional laboratory methods when it comes to detection limits. Light elements (atomic number <16) are difficult to identify by XRF because of the absorption of their fluorescent X-rays. A number of other effects further impede the general use of XRF [1,9]. Data can be unreliable if the soil moisture content is above 20 wt % [14], if the soil particles are large and not uniformly distributed within the sample, and if the surface condition is rough and heterogeneous [10]. When fluorescent X-rays are absorbed by another atom within a sample on their way to the detector, secondary fluorescence is generated and the XRF spectrum is altered [15]. Rayleigh and Compton scattering further hamper analysis when X-rays from the tube anode reach the detector. In addition, mathematical corrections within the instrument's software are needed to resolve peak overlaps on the XRF spectrum between different elements. Lastly, sum peaks occur when two fluorescent photons reach the detector simultaneously, and escape peaks appear when fluorescent X-rays are partly absorbed by the detector.

Despite these drawbacks, elemental analysis by XRF has proven to correlate well with commonly used laboratory methods such as atomic absorption spectrometry (AAS), inductively coupled plasma atomic emission spectrometry (ICP-AES), and inductively coupled plasma mass spectrometry (ICP-MS) [7,16,17]. Yet, soil datasets in these comparative studies were often limited to a few samples (<50) [11,18,19], a few elements (1–2) [20–24], or to a restricted study area with similar land use or similar parent material [11,13,21,25,26]. Additionally, only a few studies have investigated the impact of different XRF scanners by comparing their mutual performances [27,28].

This study compares the precision and general performance of three XRF scanners to those obtained by conventional laboratory analysis, using a soil database more extensive than in previous studies, including 15 elements and 128 samples gathered from divergent land managements from different countries in Europe, Africa and the Arabian Peninsula. Next to evaluating the general applicability of XRF for soil elemental analysis, the effect of multiple soil characteristics on the XRF performance is investigated. By evaluating the accuracy of XRF for soil elemental analysis in a controlled laboratory environment, our aim is to provide fundamental insight into reliability of such instruments. Such background knowledge is crucial to further develop applications in the field where less predictable conditions will exacerbate XRF reliability problems. Since XRF is being used increasingly as a fast and easy elemental analysis tool by end-users who lack expert knowledge needed to fine-tune XRF scanners, the latter were used with a standard soil calibration provided by the supplier, and all scanners were employed in benchtop mode to ensure a controlled experimental set-up.

## 2. Materials and Methods

### 2.1. Soil Sample Collection and Preparation

A total of 128 homogenized soil samples originating from 10 different countries were considered in this study. They represented different textural classes and land managements, resulting in a very diverse soil database (Figure 1) [29–36]. An overview of the samples along with their specific soil characteristics and elemental concentrations can be found in the Supplementary Materials. The samples were air-dried, sieved through a 2 mm stainless steel sieve, grinded and intensely mixed. Then, 30 mm open-ended XRF cups (Malvern Panalytical Ltd., UK) were filled with 2 g of soil and fixed with an additional cotton ball. The cups were sealed with a 4.0-μm prolene X-ray film (Chemplex Industries, USA).

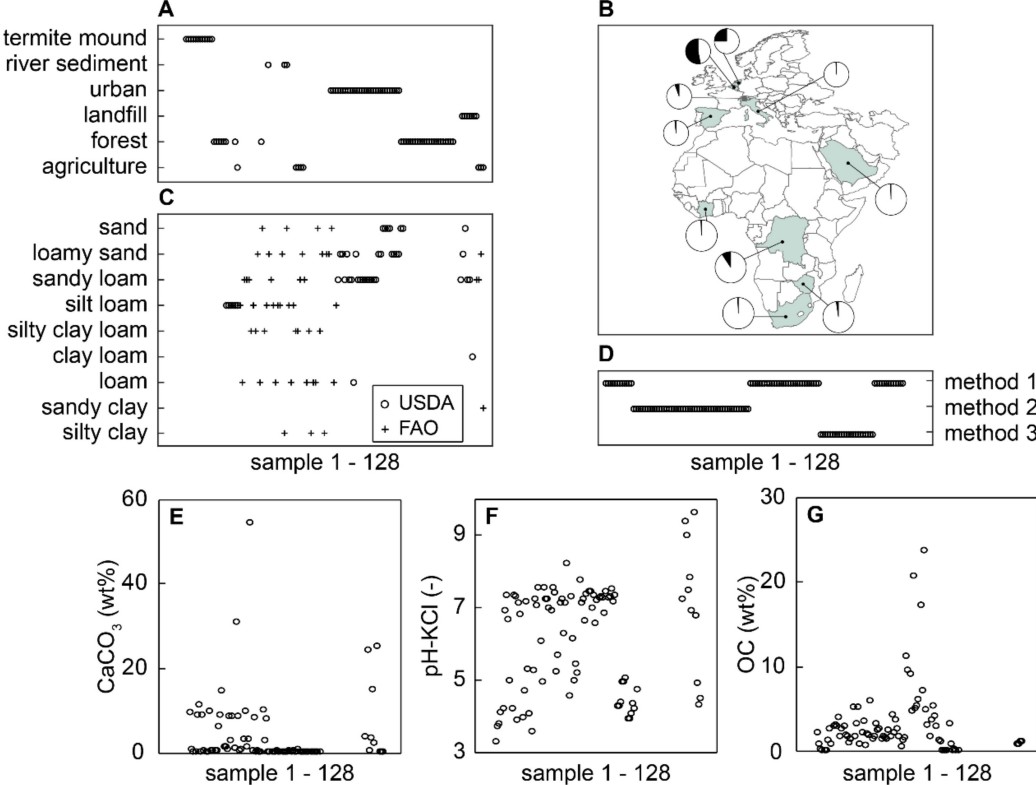

**Figure 1.** Soil sample database. Overview of land management (**A**), country of origin (**B**), textural class (**C**), laboratory analytical method used (**D**), CaCO3 (**E**), pH-KCl (**F**) and organic carbon (OC) content (**G**) for the samples in the soil database. For some samples, part of these data is missing. The share of samples collected in each country is shown through pie charts and further details are available in the supplementary file. For details on the laboratory analytical method used, see Table 1. The textural classes are based on the USDA (o) [37] and FAO (+) [38] particle size fractions.

**Table 1.** Laboratory methods. Overview of various laboratory methods used to determine elemental concentrations of the soil database. Figure 1 shows the method used for every sample. ICP-AES = inductively coupled plasma atomic emission spectrometry.

| Method | Digestion | Solvents | Detection | Number of Samples |
|--------|-----------|----------|-----------|-------------------|
| 1 | alkaline fusion | $Li_2B_4O_7 + LiBO_2$ | ICP-AES | 61 |
| 2 | acid digestion (total) | $HNO_3 + HF$ | ICP-AES | 44 |
| 3 | acid digestion (partial) | $HNO_3 + HCl$ | ICP-AES | 23 |

## 2.2. XRF Analysis

The concentrations of 15 elements commonly included in standard XRF soil calibrations (lead (Pb), cadmium (Cd), arsenic (As), zinc (Zn), copper (Cu), nickel (Ni), iron (Fe), manganese (Mn), chromium (Cr), calcium (Ca), potassium (K), sulfur (S), phosphorus (P), aluminum (Al) and magnesium (Mg)) were measured in each sample by three XRF scanners, in which X-rays were emitted from a Rh tube and detected on a large area silicon drift detector (SDD). Following scanners and built-in calibrations as provided by the supplier were employed: a Vanta VMR M Series portable scanner (Olympus, Germany) using the Geochem calibration, a S1 Titan 800 scanner (Bruker, USA) using the DualSoil calibration and a XMET 8000 Geo Expert handheld XRF analyser (Oxford Instruments, UK) using the Soil calibration. S, P, Al and Mg concentrations cannot be measured using this latter calibration, and due to time constraints, a subset of only 56 randomly chosen samples was additionally measured using the built-in mining light elements (LE) fundamental parameters (FP) calibration. As this paper aims to assess the robustness of XRF in general, but does not intend to designate the best scanner,

only blinded results will be shown. The experimental set-up was similar for every employed scanner. Each sample was measured in two phases: heavy elements (Pb, Cd, As, Zn, Cu, Ni, Fe, Mn) and light elements (Cr, Ca, K, S, P, Al and Mg) were measured at high voltage (40–45 kV) and at low voltage (8–15 kV), respectively. Note that, although Cr, Ca and K (atomic number > 16) are not regarded as light elements in this study, built-in soil calibrations did measure these elements at a low voltage. Each phase was implemented during 60 seconds to ensure high precision of the results. The scanners were used in benchtop mode and, for each of them, the samples were measured three times by moving the sample cups over the measurement window of the scanner. The three records were then averaged to obtain final elemental concentrations for each sample and each scanner. Due to time constraints, the 56 samples analysed with the XMET 8000 Geo Expert scanner using the Mining calibration were measured only once.

*2.3. Laboratory Analysis*

The elemental concentrations of all 128 samples were determined on remaining soil material by conventional laboratory analysis using different methods (Table 1, Figure 1). Variation between the methods is considered negligible compared to the variation between the XRF and the analytical results. Methods 1 [39] and 2 [40] measure real total elemental concentrations, whereas the aqua regia digestion used in method 3 [41] does not dissolve silicates, iron oxides and associated metals [42]. As such, the complete removal of some elements from the soil exchange complex is prohibited. However, the obtained pseudo-total concentrations do not tend to deviate much from the real total concentrations for Cd, Cr, Cu, Ni, Pb and Zn. Therefore, for the 23 samples analysed by method 3, real total concentrations of these elements were calculated based on regression curves between real total and pseudo-total concentrations, established by Seuntjens et al. (2006) [43]. The Al, Fe, Mn, As, Ca, K, Mg, P and S concentrations measured with Method 3 were disregarded as these pseudo-total concentrations likely deviated too much from real total concentrations.

*2.4. Statistical Analysis*

Skewness and kurtosis tests revealed that, for most elements and scanners, the data were not normally distributed [44]. Results have therefore been analysed using non-parametric statistics. For every element and each XRF scanner, scatterplots were made showing the XRF concentrations as a function of the analytical concentrations. Corresponding Spearman correlation coefficients ($r_s$) were calculated and compared to the following data quality limits as adapted from the United States Environmental Protection Agency (USEPA) [9]: A definitive data level quality is obtained for elements where $r_s \geq 0.9$ and where XRF concentrations equal laboratory concentrations based on a Wilcoxon rank sum test (1% significance level). Screening data level quality is obtained for elements where $r_s \geq 0.7$ and poor data level quality is obtained for elements where $r_s < 0.7$. The recovery rate for each scanner (j = 1, 2, 3) and each element (i = 1, 2, ..., 15) was calculated based on the 128 (N) samples using formula 1.

$$\text{recovery rate}_{i,j} \, (\%) \; = \frac{1}{N} \sum_{n=1}^{N} \frac{\text{XRF concentration}_{n,i,j}}{\text{laboratory concentration}_{n,i}} \times 100. \tag{1}$$

Boxplots were made showing the distribution of the soil elemental concentrations as measured by laboratory analysis and XRF scanners 1, 2 and 3. For each element, the variability between these data distributions was analyzed through a Kruskal-Wallis test (5% significance level).

For each sample (n), the scaled dissimilarity was calculated as

$$\text{scaled dissimilarity}_{n,i,j} \, (-) \; = \frac{\text{XRF concentration}_{n,i,j} - \text{laboratory concentration}_{n,i}}{\max\left(\text{laboratory concentration}_i\right) - \min\left(\text{laboratory concentration}_i\right)}, \tag{2}$$

where i is the respective element (i = 1, 2, . . . , 15) and j the respective scanner (j = 1, 2, 3). The effect of commonly available soil characteristics (organic carbon (OC) content, CaCO3 content, clay content and pH-KCl) on the XRF performance for each element was analysed by plotting the scaled dissimilarities as a function of the samples' soil characteristics and calculating the corresponding Spearman correlations. Next, the calculated scaled dissimilarities were averaged for each scanner and each element, and mean values were plotted against the elements' atomic masses. Linear regression analysis was then applied to evaluate whether or not the XRF scanners were equally sensitive to the atomic weight of the elements.

## 3. Results

### 3.1. Scatterplots of Elemental Concentrations Measured by XRF Versus Conventional Chemical Analysis

Scatterplots of the XRF results against the analytical results show that a one-to-one linear relationship was obtained for most elements (Figure 2). Although the limits of detection (LOD) for laboratory analysis and XRF were rather low for Cd (<20 ppm), the element was only detected by laboratory analysis in 44 out of 128 samples, and by XRF in 2, 4 and 3 samples for scanners 1, 2 and 3, respectively. Due to this lack of data, only three sample results could be shown for Cd. As (arsenic) was only detected by laboratory methods in 43 samples (LOD ≤ 20 ppm), whereas XRF results were obtained in 123, 106 and 89 samples for scanners 1, 2 and 3, respectively, enabling to plot the results of 43, 43 and 37 samples, respectively. The scatterplots revealed; that all XRF scanners overestimated the analytical concentrations in some samples for Cu, S, P and Mg (Figure 2). Ca and Al concentrations were only overestimated by scanner 1, and the K concentration of two samples analyzed by scanner 3 proved to be strongly underestimated.

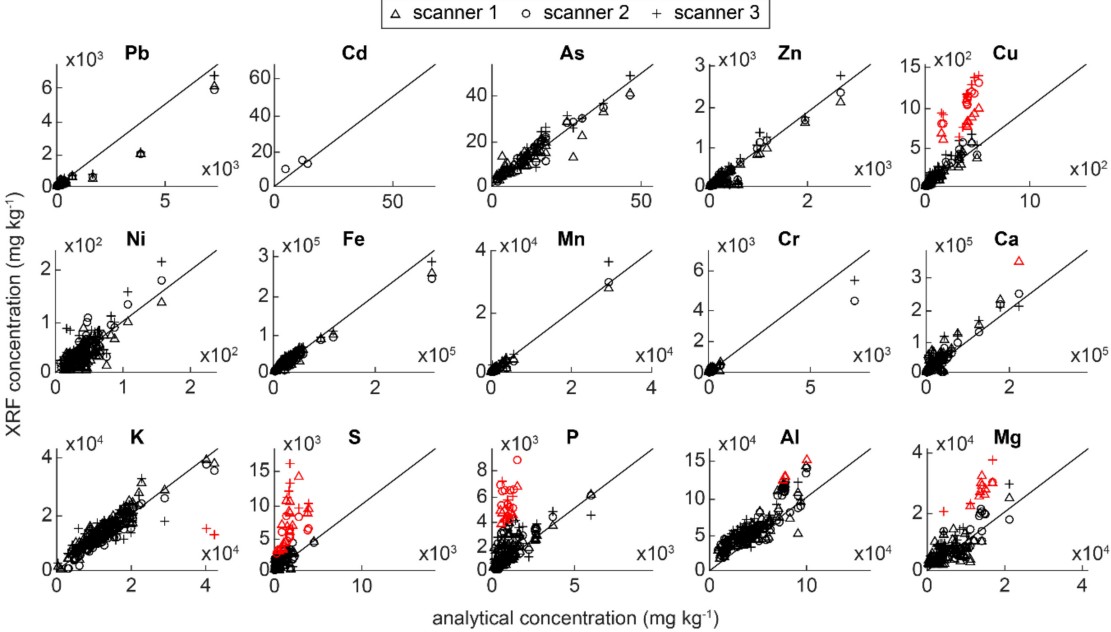

**Figure 2.** Scatterplots of X-ray fluorescence (XRF) versus laboratory results. Scatterplots of the element concentrations measured by XRF compared to laboratory analysis for each element and each tested XRF scanner. The 1:1-line is represented in black. Overestimated and underestimated samples for which abs[scaled dissimilarity] > 0.5 are shown in red.

### 3.2. Correlation Coefficients and Recovery Rates

Significant (5% significance level) positive correlations were obtained between the XRF and analytical concentrations for all elements except Cd (Figure 3a). According to adapted USEPA data quality levels [9], XRF data quality commonly reached screening level for most elements except Ni, Cr, S and P, for which only some scanners reached screening level (Table 2). A definitive data quality level

was only obtained for Mn, Ca and Al, but this might partly be due to the low power of non-parametric tests. The calculated recovery rates reveal that the concentration of most elements was overrated by XRF (Figure 3b). Although the XRF performances for Ni, Mn, Cr, Ca, K, S and Mg were highly dependent on the scanner, worse results were generally obtained for lighter elements, with better results for Al and Mg than for S and P.

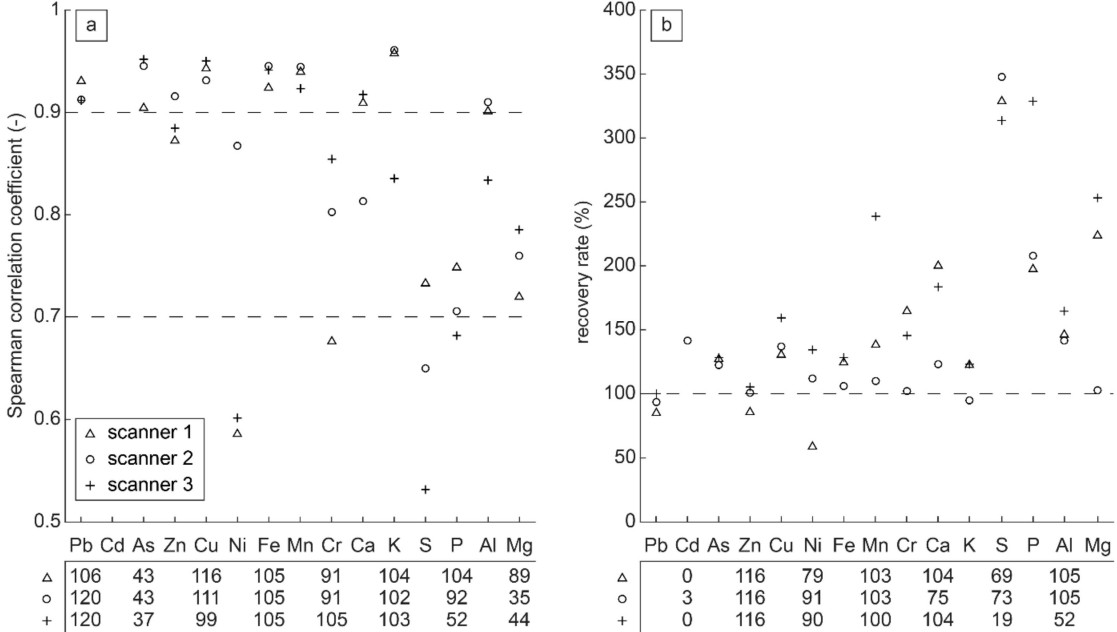

**Figure 3.** Calculated statistics. (**a**) Spearman correlation coefficients; and (**b**) recovery rates for each element and each X-ray fluorescence (XRF) scanner. The atomic number (Z) of the elements decreases from left to right. Due to a lack of data, results could not be plotted for Cd measured with scanner 1 and 3. No significant correlation ($p < 0.05$) was found for Cd with scanner 2 as the element was only detected in four samples. The number of soil samples used to calculate the correlations and recovery rates is given below the figures for each element and each scanner.

**Table 2.** Data quality levels. X-ray fluorescence (XRF) data level quality for each scanner and each element based on significant ($p < 0.05$) Spearman correlation coefficients between XRF and laboratory concentrations according to USEPA [9]. x = definitive data level quality, o = screening data level quality, - = poor data level quality ($r_s < 0.7$) and n.s. = not significant.

| Element | Atomic Number (Z) | Scanner 1 | Scanner 2 | Scanner 3 |
|---------|-------------------|-----------|-----------|-----------|
| Pb | 82 | o | o | o |
| Cd | 48 | n.s. | n.s. | n.s. |
| As | 33 | o | o | o |
| Zn | 30 | o | o | o |
| Cu | 29 | o | o | o |
| Ni | 28 | - | o | - |
| Fe | 26 | o | o | o |
| Mn | 25 | o | o | x |
| Cr | 24 | - | o | o |
| Ca | 20 | x | o | o |
| K | 19 | o | o | o |
| S | 16 | o | - | - |
| P | 15 | o | o | - |
| Al | 13 | x | x | o |
| Mg | 12 | o | o | o |

### 3.3. Comparison of XRF Scanners

Figure 4 shows the boxplots of the analytical and XRF concentrations for all elements. A Kruskal-Wallis test was applied to all data, including outliers. The test showed no significant differences between the laboratory methods and all three scanners for Pb, As, Fe and Ca. Soil elemental concentrations measured in the laboratory differed significantly from those measured by scanner 1 for Zn, Ni, Cr, S and P. The analytical concentrations also differed from those measured by scanner 2 for Cd, S, P and Mg, and measured by scanner 3 for Cd, Cu, Ni, Mn, S, P, Al and Mg. Elemental concentrations generated by scanner 1 differed from those provided by scanner 2 for Ni, Cr, K and Mg, and from those by scanner 3 for Cu, Ni, Mn, P, Al and Mg. Lastly, scanners 2 and 3 provided different measurements for Mn, K, P and Al. Significant differences have been shown for Cd, Zn, Cu, Ni, Mn, Cr, K, S, P, Al and Mg, but these might partly be due to the use of non-parametric tests and small sample sets (e.g., Cd was only detected in 2, 4 and 3 samples by scanners 1, 2 and 3, respectively) (Figure 4). Linear regression analysis further suggested that scanner 3 produced poorer results than scanner 1 and 2 for lighter elements (Figure 5). However, the average scaled dissimilarity for S measured with scanner 3 was 1.14, and a high RMSE value for the linear regression (0.29) confirms that this outlying value biased the regression line. RMSE values for scanner 1 and 2 were only 0.11 and 0.10, respectively.

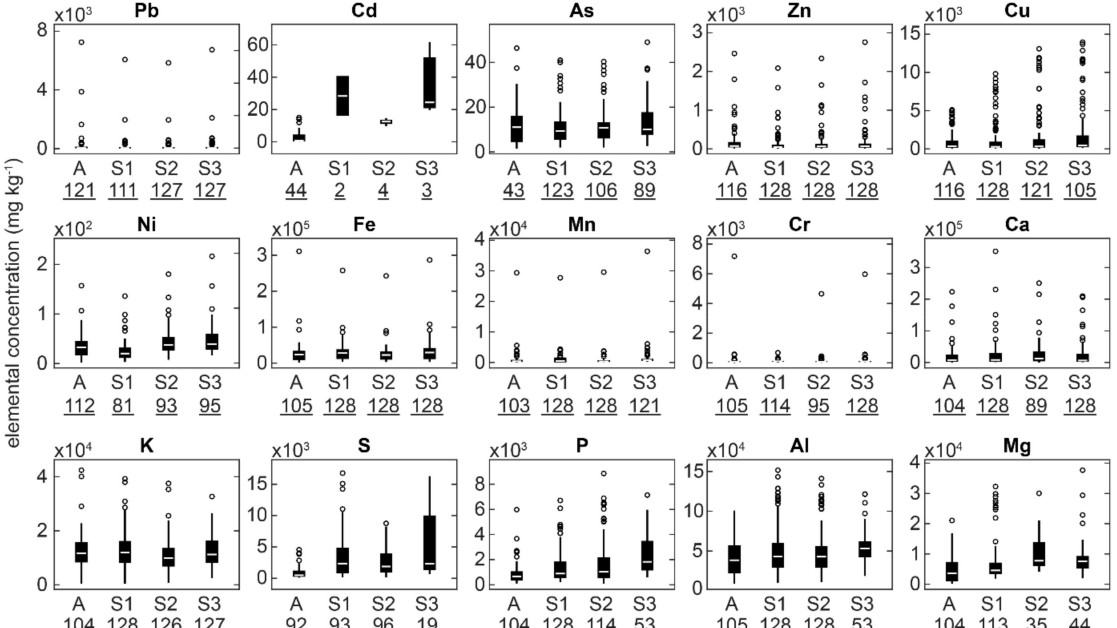

**Figure 4.** Boxplots of soil elemental concentrations. Boxplots showing the distribution of elemental soil concentrations in all samples as measured by laboratory analysis (A), scanner 1 (S1), scanner 2 (S2) and scanner 3 (S3). The white line corresponds to the median value, the whiskers correspond to the most extreme data points which are not outliers and the dots correspond to extreme values considered as outliers (>1.5 × interquartile range). The number of soil samples comprised within each boxplot is written underneath each boxplot (underlined).

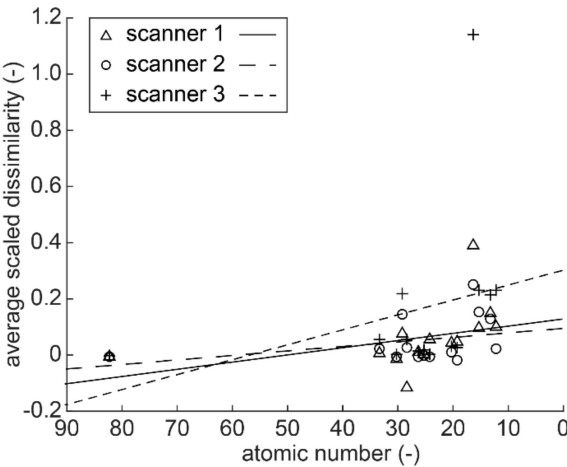

**Figure 5.** Sensitivity of X-ray fluorescence (XRF) to the atomic weight of the elements. Relationship between the elements' atomic numbers (Z) and the corresponding average scaled dissimilarities for the three tested scanners. A least-squares linear regression line is shown for each scanner. Results for Cd were left out of the regression analysis as the lack of data for scanner 1 and 3 would give biased results.

### 3.4. Effect of Soil Characteristics on XRF Performance

Scaled dissimilarities were calculated for all samples and plotted against selected soil characteristics (Figures 6–9). Small disparities between XRF and analytical concentrations were reflected in scaled dissimilarities approaching to zero (<0.2), whereas larger values depicted substantial differences between both concentrations. Elements sensitive to over- or underestimation by XRF (Figure 2) showed the largest absolute scaled dissimilarities. However, the scaled dissimilarities did not seem dependent on the selected soil characteristics (Figures 6–9). Although correlations between the dissimilarities and the soil properties reached significant absolute values between 0.21 and 0.68, the uneven distribution of soil characteristics within the data set prevented making robust conclusions about the general effect of soil characteristics on XRF performance. Too few samples with a high clay content, high OC content, high CaCO3 content and a non-neutral pH-KCl were present in the data set. The scaled dissimilarities generally approached zero for the whole range of observed OC contents, CaCO3 contents, clay contents and pH-KCl values, but increased (absolute) dissimilarities were obtained for soil characteristic values frequently appearing in many samples.

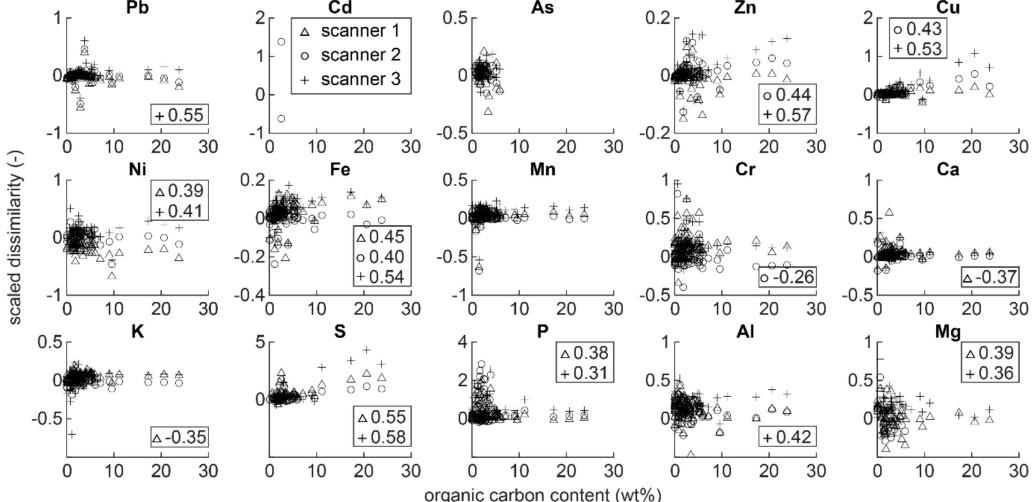

**Figure 6.** Effect of organic carbon (OC) on X-ray fluorescence (XRF) performance. Scatterplots of the scaled dissimilarities for all samples and all scanners versus the samples' OC contents. Significant (5% significance level) Spearman correlations between the scaled dissimilarities and the OC contents are shown for each scanner.

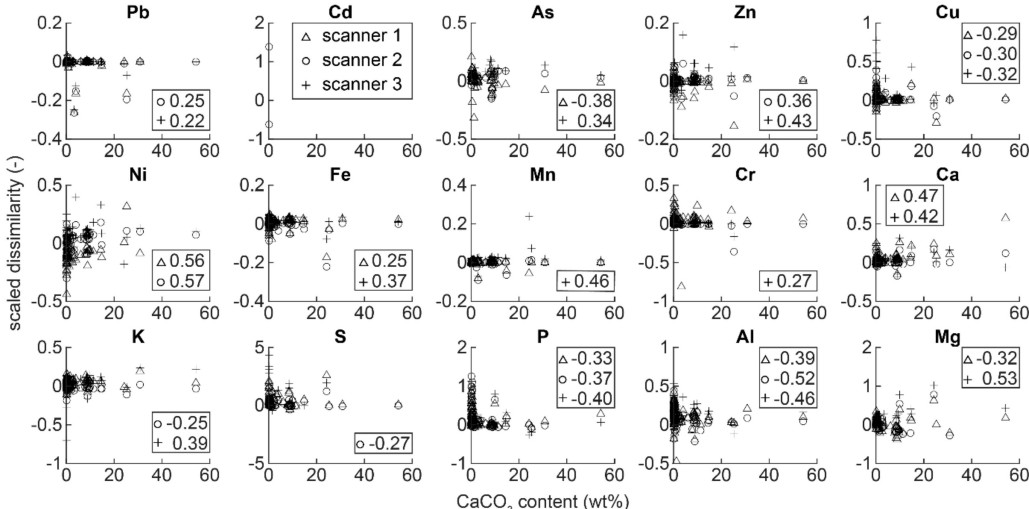

**Figure 7.** Effect of CaCO$_3$ on X-ray fluorescence (XRF) performance. Scatterplots of the scaled dissimilarities for all samples and all scanners versus the samples' CaCO$_3$ contents. Significant (5% significance level) Spearman correlations between the scaled dissimilarities and the CaCO$_3$ contents are shown for each scanner.

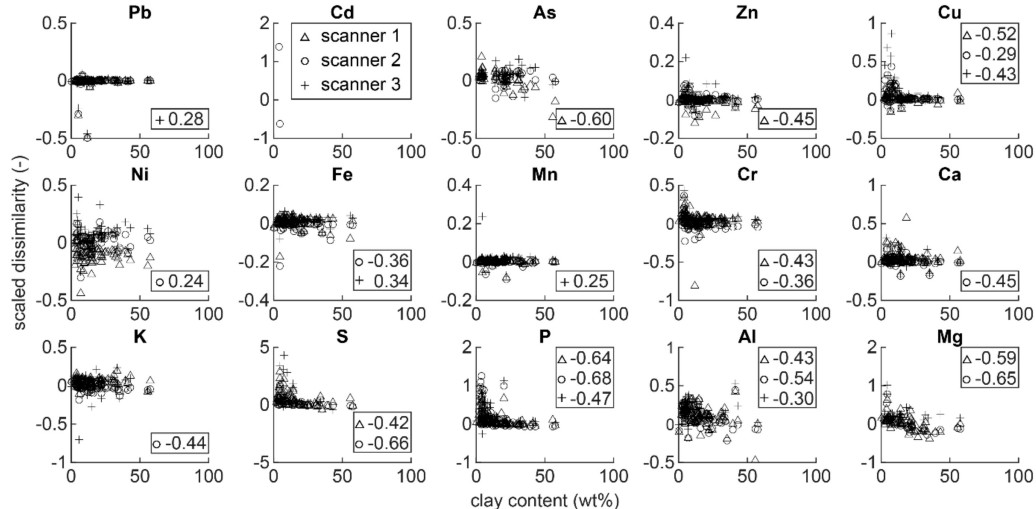

**Figure 8.** Effect of clay content on X-ray fluorescence (XRF) performance. Scatterplots of the scaled dissimilarities for all samples and all scanners versus the samples' clay contents. Significant (5% significance level) Spearman correlations between the scaled dissimilarities and the clay contents are shown for each scanner.

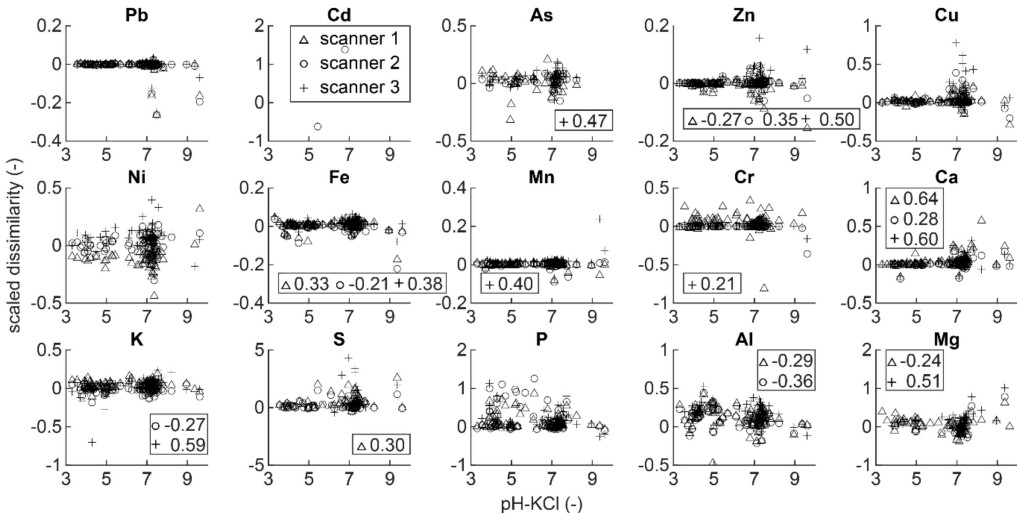

**Figure 9.** Effect of pH-KCl on X-ray fluorescence (XRF) performance. Scatterplots of the scaled dissimilarities for all samples and all scanners versus the samples' pH-KCl values. Significant (5% significance level) Spearman correlations between the scaled dissimilarities and the pH-KCl are shown for each scanner.

## 4. Discussion

### 4.1. Overestimation of Elemental Concentration

Compared to the chemical analyses, XRF overestimated the concentration of Cu, S, P and Mg in several samples from different origins (Figure 2). Ca and Al concentrations were only overestimated by scanner 1 in 1 and 5 samples, respectively. The provenance variation of the overestimated samples suggests that such overestimations are independent of the soil matrix. High elemental concentrations, on the other hand, consistently led to overestimation of Cu, S and Mg. Only for P, the highest concentrations were properly estimated by XRF. The ranges of the training data set used to create the built-in soil calibrations might not have been equivalent to the ranges of our data set, leading to biased results when extrapolating beyond the ranges of the training data set. On the other hand, our data might not respond well to the linear relationships applied in the built-in soil calibrations. Supposing, for instance, our data follow a logarithmic relationship, increasingly worse results would be obtained for samples with increasing elemental concentrations when applying a linear relationship to the data set. In any case, care should be taken when using the built-in calibrations of the evaluated instruments for estimating expected high concentrations of Cu, S and Mg. However, empirical recalibration using validation samples specific to a particular dataset or development of improved calibrations might overcome this drawback [45].

### 4.2. Comparison of Different XRF Scanners

Although the calculated recovery rates show that the concentration of most elements is overestimated by XRF (Figure 3b), in general, good results were obtained for all scanners. In 86% of the cases, a screening data level quality was obtained (Table 2). Scanner 2 provided the best overall results, whereas the poorest results were obtained for scanner 3 (Figure 3). A definitive data level quality was obtained by scanner 1 for Ca and Al, whereas scanners 2 and 3 only properly estimated Al and Mn, respectively. Screening data level quality was obtained by scanner 2 for all elements except S, whereas scanner 1 failed for Ni and Cr, and scanner 3 failed for Ni, S and P. In addition, scanner 3 seemed to be insensitive to high K concentrations (Figure 2). The varying performance of the investigated scanners is confirmed by the variability obtained between the boxplots (Figure 4) and can partly be attributed to the different built-in soil calibrations and corresponding instrument settings (e.g., voltage) used, though also other technical aspects can influence their performance (e.g., detector sensitivity, filters, software for resolving peak overlaps, etc.). In this

research, scanner 2 performed best under default conditions frequently used by non-experts. However, since instrument settings (e.g., voltage and exposure time) and calibrations can be adjusted, it remains undecided whether this scanner performs best in other settings or for broader applications. Empirical recalibration to one's own needs would improve the XRF performance as it would correct for site-specific matrix effects [8,9,27]. Adjusting the intercept and slope of the standard soil calibrations might reduce the absolute differences between XRF and laboratory results while maintaining high correlations between both.

### 4.3. XRF Performance for Light Elements

The XRF performance generally decreased with decreasing atomic weight (Figures 3 and 5). In addition, scanner 3 seemed to produce worse results for lighter elements compared to scanners 1 and 2, but the regression line of scanner 3 might be biased due to a large amount of overestimated S concentrations (Figure 5). For light elements, the limited amount of energy captured within the emitted X-rays is easily absorbed in the soil matrix and ambient air. The detector does not receive sufficient amounts of energy to distinguish significant concentrations of a certain element. In addition, background scatter tends to dominate the low energy peaks of light elements, making them hardly recognizable in an XRF spectrum [46,47]. While these drawbacks were mitigated in all scanners by large area silicon drift detectors (SDD), which improved detection of light elements, correlations and recovery rates for P, S, Al and Mg (atomic weight ≤16) were generally worse than for heavier elements. Better results were obtained for Al ($Z = 13$) and Mg ($Z = 12$) than for S ($Z = 16$) and P ($Z = 15$) (Figure 3, Table 2). Firstly, Al and Mg tend to be more efficiently detected by all XRF scanners since these elements are naturally present in soils in higher amounts than S and P (Figure 2). Secondly, the training data sets used to build the soil calibrations might have been more equivalent to our data set for Al and Mg than for S and P. But lastly and most importantly, the X-rays were distributed by an Rh tube, known to give better results for Al and Mg [12]. Cd concentrations, on the other hand, were very low in almost all samples and hardly detected by the XRF scanners because the Rh tube is not well suited for measuring Cd (Figure 2) [12].

### 4.4. Effect of Soil Characteristics on XRF Performance

High soil moisture contents and the presence of large, rough soil particles within a sample impede a proper estimation of elemental soil concentrations [10,14,16,48–50]. In this study, samples were dried and grinded to circumvent these drawbacks. The effect of moisture content and particle size on the XRF performance could therefore not be evaluated, but the influence of four other commonly available soil characteristics was investigated (Figures 6–9). However, due to an uneven distribution of OC contents, CaCO3 contents, clay contents and pH-KCl values within the sample suite, the data set was not fully suited to fulfill this goal. No robust conclusions regarding the effect of soil characteristics on the XRF performance could be made and the results are rather indicative. It was expected that high OC contents in the samples would dilute the elemental concentrations because XRF fails to accurately measure light elements such as C, H, N and O [1,9,51,52]. A similar effect was hypothesized for CaCO3, albeit less pronounced than for OC since Ca is better detected by XRF than C, H, N and O. In addition, low CaCO3 contents and low pH-KCl values often inherent to sandy soils were expected to increase the scaled dissimilarities as larger, sand-sized particles give worse XRF results than smaller, clay-sized particles [48]. Likewise, samples with a low clay content were expected to also increase dissimilarities. However, scaled dissimilarities generally approached zero for the whole range of soil properties (Figures 6–9), suggesting such effects were absent in our results. The instrument sensitivity appeared to be similar for the entire range of OC, CaCO3 and clay contents, and pH-KCl values in analysed samples.

## 5. Conclusions

Three commercially available XRF scanners were evaluated for the rapid measurement of the most commonly demanded soil elements in a suite of samples sourced from widely different

environments. Elemental concentrations measured by the three XRF scanners were mutually compared with each other, and with results obtained from conventional chemical methods. Although all scanners slightly overestimated analytical concentrations, the instruments produced acceptable results even for lighter elements, especially Al and Mg. Yet, Cu, S and Mg were not properly measured in high concentrations—Ni, Cr, S and P did not reach screening level; and high accuracy was obtained for Mn, Ca and Al for some scanners. Unexpectedly, XRF performance did not seem to depend on specific soil characteristics such as high OC or CaCO3 contents. Our results showed that XRF scanners applied with standard soil calibrations can be reliably used in lab environments by non-experts for measuring soil geochemistry. An accurate estimation of soil elemental concentrations might, however, be impeded when using these instruments in the field (i.e., in portable XRF (pXRF) mode), where high soil moisture contents and a lack of sample preparation could complicate robust elemental analysis [50,53]. This study can thus be considered a best-case scenario, demonstrating the potential of soil elemental analysis with XRF in a fully controlled laboratory environment. As such, it contributes to setting basic background knowledge needed to develop XRF applications in the field. Studies quantifying the effect of soil moisture content, soil heterogeneity, and soil texture on XRF performance could further aid the design of robust, portable XRF scanners. However, XRF performance does hinge on factors such as customer usability and the instruments' capacity to correct for peak overlaps. Although built-in soil calibrations allow non-experts to readily analyze soil samples, it is apparent that empirical recalibration to fit case-specific needs would render more reliable results. However, such recalibration can be time-consuming and requires expert knowledge alongside access to appropriate laboratory facilities.

**Supplementary Materials:** The following are available online at http://www.mdpi.com/2072-4292/11/21/2490/s1. XRF and analytical elemental concentration data are provided in a supplementary spreadsheet file (data.xlsx). S, P, Al and Mg concentrations were only measured in 56 samples with the XMET 8000 Geo Expert handheld XRF analyser (Oxford Instruments, UK), and to preserve the anonymity of the scanners, results for these elements are omitted from the supplementary file.

**Author Contributions:** Conceptualization, P.D.S., P.F., A.M.M., M.V.M. and A.V.; Investigation, Y.D., N.D., J.D.G. and D.V.; Writing—Original Draft Preparation, Y.D. and N.D.; Writing—Review & Editing, Y.D., N.D., J.D.G., P.D.S., P.F., A.M.M., S.N., D.V., M.V.M. and A.V.; Supervision, P.D.S. and A.V.

**Funding:** This research was funded by the Research Foundation Flanders (FWO), grant numbers 1S33417N and 1288117N.

**Acknowledgments:** We would like to acknowledge the contribution of the COST Action SAGA: The Soil Science & Archaeo-Geophysics Alliance-CA17131 (www.saga-cost.eu), supported by COST (European Cooperation in Science and Technology). We thank De Looper Analytical B.V., VM ViSion International bvba and Benelux Scientific bvba for providing the XRF scanners.

**Conflicts of Interest:** The authors declare no conflict of interest.

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
