# Peer review of "A Comprehensive Study of Three Different Portable XRF Scanners to Assess the Soil Geochemistry of An Extensive Sample Dataset"

_remotesensing, doi:10.3390/rs11212490_

Round 1

Reviewer 1 Report

The manuscript reports a comparison of three X-ray fluorescence scanners to detect the feasibility of using these technologies for fast soil analyses. The paper highlights an interesting gap: to move to massive use of X-ray fluorescence technologies for soil analysis.

The introduction is based on solid research on previous work. The methodology applied is well described and based on appropriate statistical tests. The results are clearly described and supported by visual plots. The interpretation of the results is brief but reasonable.

Here are some specific comments:

Lines 148-149: The sentence “Few data points were also available for As” is not consistent with the scatter plot in Fig. 2

Fig. 2: Cd graph is not clearly visible because of the legend.

Reviewer 2 Report

GENERAL COMMENTS

The paper is solid, valuable and worth to be published in international journal after some corrections.

On the other hand, I have some doubts, if this paper is particularly suitable for Remote Sensing journal. In my opinion, a term of remote sensing refers to acquisition of information about objects studied from distance. Actually, the authors of present paper compare laboratory analysis using different (!?) methods with readings of different sensors made also in laboratory conditions, after drying, sieving and grinding of soil samples. For this reason I think, that the paper explores rather the possibility of replacing expensive and invasive methods of laboratory analysis with XRF sensors and may be useful in designing of small and cheap laboratories. But, at this moment the paper does not provide many information directly related to remote sensing area itself.

For this reason I suggest 2 possibilities:

1) Submitting the paper to other journal, more related to the topic of this paper, e.g. Sensors, Chemosensors etc… after some minor corrections (see detailed comments);

2) Accepting the paper for publication in Remote Sensing journal after completing it with some investigations more directly related with the area of remote sensing, for example by additional studies including (major revision):

a) the use of XRF scanners on sieved and ground samples, but with variable, controlled humidity – thus, the study will provide valuable information for the interpretation of field, remotely obtained data, and correction of it depending on humidity;

b) the use of XRF scanners on the dry, but not sieved and ground samples – such study would simulate the remote sensing in dry conditions.

The perfect study would include the XRF scanning in field conditions, then in the samples before drying and sieving, determination of moisture, and then the study similar to described in the present paper. Although, the laboratory analysis should be performed with the same method of all samples. Anyway, the perfect studies do not exist and this study should be published, although I wonder, if the Remote Sensing is the most suitable journal for this paper in current form.

Additionally, I have some detailed suggestions regarding data presentation and processing. Although they seem numerous, in my opinion are not very difficult to be made. Some of my comments may be useful in preparing other paper based on the same data.

DETAILED COMMENTS

Line 61. In my opinion, this database is not very extensive, although it may be much more extensive, than the databases used in previous studies. I suggest to replace this phrase by “...using database more extensive than in earlier studies, including 128 soil or sediment samples”, or similar. It may be also useful adding information on number of samples and the elements considered in previous studies. It will highlight the value of current study.

Lines 68-74 – 2.1. Soil samples collection and preparation. I think, it would be useful to provide additional information regarding soil parent material (aeolian, glacial, aluvial and others perhaps epoch of origin), type of material (mineral soil, organic soil, sediment etc.) and to add, e.g in supplementary file. Such information, if available, could be also to verify, if these factors were related to performance of particular XRF scanners.

Lines 94-95. Here, the authors should clarify, which is the limit of atomic number separating light and heavy elements. In the version I received for review, this limit is 24 (chromium, which was treated as light element in this study), however it is inconsistent with the statement in Introduction (line 42), where atomic number 16 is mentioned as respective limit in one of previous studies. This is acceptable, but the authors should clarify it, to avoid confusion during the reading of the paper.

Line 100 – why 56 samples analysed with XMET 8000 were measured only once? Please, add a short explanation.

Lines 102-117 – 2.3. Laboratory analysis. The use of three different methods in laboratory analyses consists a drawback of this study, although it is, most probably, related with the extension of database originated in different countries. However, almost all the statements present in this study should be supported by referring to adequate, published sources about these methods of analyses, or to own data of authors, which are shown in supplementary file. I think, such references should be added in lines 105, 107, 110 and 114. When referring to own data, the authors should mention the name of particular sheet (e.g. KatangaRT etc...)

Line 122. The designation of Spearman’s rank coefficient with r is acceptable, but in my opinion s or rs is better, because r is most frequently used for designation of Pearson’s correlation coefficient.

Line 135. Why the sand content, instead of clay, was not considered? See the authors comment in Introduction, line 44.

Lines 118-141: 2.4. Statistical analysis. I think the authors could try to apply multiple regression (stepwise) to explain the effect of various factors (OC, pH, clay or better sand) on performance of particular scanners, and XRF in general? It would very interesting, if this is the first paper considering so extensive (although, still not very extensive in my opinion) dataset, and very useful for future studies.

Instead, it may be useful to present simple scatterplots showing relationship between scanners performance for measurement of each element (Y axis) and the factors mentioned above and other, qualitative factors (X axis: OC, pH, clay or better sand country, land use, soil texture and geology, if available).

Line 164. I think, the authors refer to the Table 2 which should be mentioned here.

Lines 167-168. This may suggest, that XRF method is more suitable for metals (Al and Mg) than for nonmetals (P and S)? If this true, this could be mentioned in discussion (4.3. XRF performance for light elements) and conclusions, especially if supported by other studies.

Lines 312-326: Conclusions. I would add, or even shortly repeat from results and discussion, some information regarding:

-overestimation of measurements of almost all elements by all scanners;

-elements, which are most and least reliably measured by XRF scanners

Reviewer 3 Report

Please find the attached

Round 2

Reviewer 2 Report

I believe that the current version of the article can be published in Remote Sensing, although some some minor corrections are still needed.

In my opinion, the main advantage and novelty of the paper consists on use of relatively extensive dataset of samples from different places of Europe, Africa and the Arabian Peninsula for assessment of the content of significant number of elements and using different portable XRF scanners.

The use of available samples, remaining, as I suppose, from other, previous studies, is also an advantage of this paper, as it allows to extend existing knowledge without carrying completely new and expensive studies.

However, such approach resulted in certain limitations of the paper, related to the combination in one dataset of results of three different methods of laboratory analysis and use of two similar, but not identical criteria for determination of soil texture classes and, as well, uneven distribution of soil characteristics. However, the authors clearly show these limitations to readers, and this permits to design more perfect, future studies.

Additionally, I have some minor suggestions for further improvement of the paper, although I do not consider them obligatory.

I suggest changing the title of the article to:

A comprehensive study of three different portable XRF scanners to assess the geochemistry of wide (extensive?) sample dataset.”

I think this title better reflects the topic of the paper. I have certain, although subjective doubts, if termite mounds may be considered as soil?

Lines 65-66: I would replace “… all over the world” with “… different countries of Europe, Africa and Arabian Peninsule” as this much better reflects the territorial extension of the study.

Lines 87-88: I would complete the phrase “The amount of samples collected in each country is shown through pie charts” with “and available in supplementary file”. Thus I suppose, that the supplementary file will available for readers, at least in part regarding sample characteristics and origin.

Lines 112-113: I suggest to remove the last phrase of this subsection, as it repeats the statement from the lines 99-102.

Line 129: I suggest to remove the acronim AAS and it’s explanation, as the method AAS was removed from the content of the table 1.

Line 160: Although the symbols of elements are widely known, exceptionally, I would put the name of element “arsenic” after it’s symbole As. My first impression, was, that the As was an adverb beginning the phrase. Although, the English native speakers may not have such impression.

Line 341: In this part of the conclusions, I would add the suggestion, which studies are needed to assure better interpretation of results of XRF scanners in the field and may be useful in designing future portable scanners. In my opinion such studies should consider the effect of soil moisture and texture, besides other factors, on measurements performed by XRF scanners.

Figures 2, 6, 7, 8 and 9 and their description: It would be interesting to check the origin and properties of the outlying samples. Perhaps, it could help to indicate which soils (sediments, materials) may be investigated with XRF scanners, and which not.

Reviewer 3 Report

It contains in various sections. Line wise comments bellow-

33: In this section, can you please justify why people read your paper
from this journal. How is you paper important for this remote sensing
and relevant research area?

78: Every country has different levels of soil nutrients and texture. I
am wondering, did you apply any calibration methods for standardizing
the collected data, mainly texture data. Some points from my previous
question:
- how did you collected data
- how did you standardize the data to get the optimum values
- can you show descriptive statistics
- there is scale issues, how did you fix that

85: Please reorganize the figure for better understanding of the
readers. You can add letters for the figures. Please add (a), (b), (c)
and described in the section what it is.
Please remove the duplicated figures.

240: I did not find any discussion for scale dissimilarities (specially
figure 7, 8 & 9) and it impact on XRF performance. For this, you need:
- need to clarify what is scale variations in your case
- what methods you applied for this
- how did you correlate these datasets
- how did you use this data to evaluate the XRF performance

253: You need more discussion on the produced figures in the results and
discussion.

327: This part is important for other researcher and readers.

Conclusion needs:
- more valuable findings. It should not be the only results what you
achieved.
- why is it valuable for other researcher
- recommendations for others
- some future research directions from your research.
